# Pincer-cobalt boosts divergent alkene carbonylation under tandem electro-thermo-catalysis

Shulei Ge[1,2,8], Zhili Cui[1,2,8], Lei Peng[3,4,8], Xintong Wang[5], Kaixin Chen[1,2], Changrui Nie[1,2], Shoucheng Dong[1,2], Yang Huang[3,4], Gen Luo [5] ✉, Lin He [3,4] ✉ & Jie Li [1,2,6,7] ✉

Catalytic multicomponent carbonylation reactions with high regio- and chemoselectivity represent one of the long-pursued goals in C1 chemistry. We herein disclose a practical cobalt-catalyzed divergent radical alkene carbonylative functionalization under 1 atm of CO at 23 °C. The leverage of the tridentate NNN-type pincer ligand is the key to avoid the formation of catalytically inert $Co^0(CO)_n$ species and overcome the occurrence of oxidative carbonylation of organozincs, selectively tuning the catalytic reactivity of cobalt center for dictating a full cobalt-catalyzed four-component carbonylation. Moreover, direct use $CO_2$ as the C1 source in the multicomponent alkene carbonylative couplings can be achieved under a tandem electro-thermo-catalysis, thus allowing us to rapidly and reliably construct unsymmetric ketones with ample scope and excellent functional group compatibility. Remarkably, our protocol encompasses a broader of polyhaloalkanes as the electrophiles, which underwent radical-relay couplings in a completely regio- and chemoselective fashion. Finally, facile modifications of drug-like molecules demonstrate the synthetic utility of this method.

Transition-metal-catalyzed carbonylation that enables straightforward access to carbogenic skeletons containing carbonyl-derived functional groups is a vital component of the synthetic toolkit, given the prevalence of such scaffolds and their widespread applications in organic synthesis, medicinal chemistry and material science[1–3]. Thus far, direct use carbon monoxide (CO) as the abundant and low-cost C1 source to the development of carbonylative synthetic methods have witnessed considerable progress during the past decades[1–5]. Whereas the noble metals (Pd, Rh, Ir, or Ru complexes) have showed good catalytic activities in various carbonylative reactions[6–9], the development of catalysts based on the naturally abundant and cost-efficient 3 d transition metals represents an attractive alternative[10–18]. Among them, the industrial friendly cobalt complexes possessing versatile potential in homogeneous catalysis[19–22], have received special recent attention. Since cobalt complexes showed important applications in the hydroformylation[23,24] and Pauson–Khand reaction[25], significant contributions have been made to further expand the cobalt-catalyzed carbonylative transformations with readily accessible starting materials. Thus far, diverse coupling reactions between the in situ formed cobalt-carbonyl intermediate **I** and N- or O-based nucleophiles have

[1]State Key Laboratory of Bioinspired Interfacial Materials Science, Soochow University, Suzhou, China. [2]College of Chemistry, Chemical Engineering and Materials Science, Soochow University, Suzhou, China. [3]State Key Laboratory of Low Carbon Catalysis and Carbon Dioxide Utilization, Lanzhou Institute of Chemical Physics (LICP), Chinese Academy of Sciences, Lanzhou, China. [4]State Key Laboratory for Oxo Synthesis and Selective Oxidation, Lanzhou Institute of Chemical Physics (LICP), Chinese Academy of Sciences, Lanzhou, China. [5]Institutes of Physical Science and Information Technology, Anhui University, Hefei, China. [6]Suzhou Key Laboratory of Pathogen Bioscience and Anti-infective Medicine, Soochow University, Suzhou, China. [7]MOE Key Laboratory of Geriatric Diseases and Immunology, Soochow University, Suzhou, China. [8]These authors contributed equally: Shulei Ge, Zhili Cui, Lei Peng. ✉e-mail: luogen@ahu.edu.cn; helin@licp.cas.cn; jjackli@suda.edu.cn

been well developed. Strategies including cobalt-catalyzed amino-carbonylative functionalization of alkenes[26–28], amino- and alkoxycarbonylation of electrophiles[29–31], as well as oxidative C–H carbonylation[32–40] have obtained significantly attentions by offering straightforward methods to the synthesis of amides and esters (Fig. 1a). However, the paucity of unsymmetric ketone synthesis via carbonylation between electrophiles and cobalt-carbonyl intermediate **II**, which formed by selective 1,1-insertion of CO with carbon-based nucleophiles, is certainly striking[10–12]. The reasons can be ascribed to the following two major challenges: i) carbon monoxide trends to coordinate with cobalt metal tightly to form catalytically inert $Co^0$-carbonyl species (**III**); ii) intermediate **II** prefers to undergo oxidative cross-coupling process to furnish symmetric ketone[41]. Therefore, the use of abundant CO gas in cobalt-catalyzed carbonylation between carbon-based nucleophiles and electrophiles for unsymmetric ketone synthesis remains a challenging topic (Fig. 1b).

In recent years, direct use of CO gas in the catalytic multicomponent carbonylative reactions (MCRs)[42–44] via radical relay pathway have emerged as an innovative strategy to rapidly synthesis value-added carbonyl compounds from simple chemical feedstocks[45–50]. Among them, the four-component alkylcarbonylation reaction of alkenes to the synthesis of unsymmetric ketones under nickel catalysis was only recently developed by Zhang[51,52]. However, due to the aforementioned issues, cobalt-catalyzed radical-relay alkene carbonylative functionalization to access versatile ketones have unfortunately thus far proven elusive. Therefore, this underdeveloped area leaves a unique chance for developing new synthetic strategy while expanding this highly rewarding scenario. To achieve this goal, we herein disclose the realization of cobalt-catalyzed divergent radical relay alkene carbonylative functionalization under 1 atm of CO at 23 °C. Indeed, the use of tridentate NNN-type pincer ligand is the key to avoid the formation of catalytically inert $Co^0(CO)_n$ and overcome the occurrence of oxidative carbonylation of organozincs, thereby selectively tuning the catalytic reactivity of cobalt for steering a rapid radical relay coupling with the in situ formed acyl-cobalt intermediate.

Notably, we have also designed a practical tandem electro-thermo-catalysis, which enables directly replace CO with $CO_2$ in this multicomponent carbonylative coupling reaction to afford a variety of functionalized ketones. Among them, a quite number of poly-halogenated electrophiles can be assigned as the radical precursors, occurring radical-relay alkene carbonylation with excellent regio- and chemoselectivity. Moreover, the salient features of our protocol include synthetic simplicity, ample substrate scope and high efficiency (Fig. 1c).

## Results

Given the important role of sulfone moieties to optimize the stability, liposolubility and metabolism of various molecules with activities of relevance to medicinal chemistry[53], we became interested in the development of cobalt-catalyzed alkene carbonylative sulfonylation with readily accessible tosyl chloride (**1a**), p-tolylzinc pivalate (**3a**) under 1 atm of CO. To achieve this multicomponent carbonylation reaction, our investigation was focused on the ligands screening (Fig. 2a). Initially, the utilization of representative bis(pyridine)s ligands **L1**–**L6**, as well as tridentate nitrogen-based ligands **L7**–**L16** only gave trace amount of the desired product **4**, while more than 90% of the symmetric ketone was detected. These results demonstrated that the addition of these ligands is inapposite to suppress the competing oxidative carbonylation of arylzincs[10,41]. Remarkably, significant breakthroughs were made by the further evaluations of other tridentate pincer-ligands of type **L17**–**L23**. Among them, 2,6-bis(N-pyrazolyl)pyridine ligand (bpp, **L17**), which has seen wide application in the nickel-catalyzed cross-coupling reactions[54,55] while rather rare application in cobalt catalysis, gave the optimal results for cobalt-catalyzed four-component carbonylation by overcoming a series of side pathways, such as oxidative carbonylation of arylzincs, sulfonylation of organozincs, alkene 1,2-arylsulfonylation, β-hydride elimination have been totally prohibited[56,57]. The desired unsymmetric ketone **4** was obtained in 75% yield under 23 °C within 2 h. Notably, switching from p-tolylzinc pivalate to other anion-

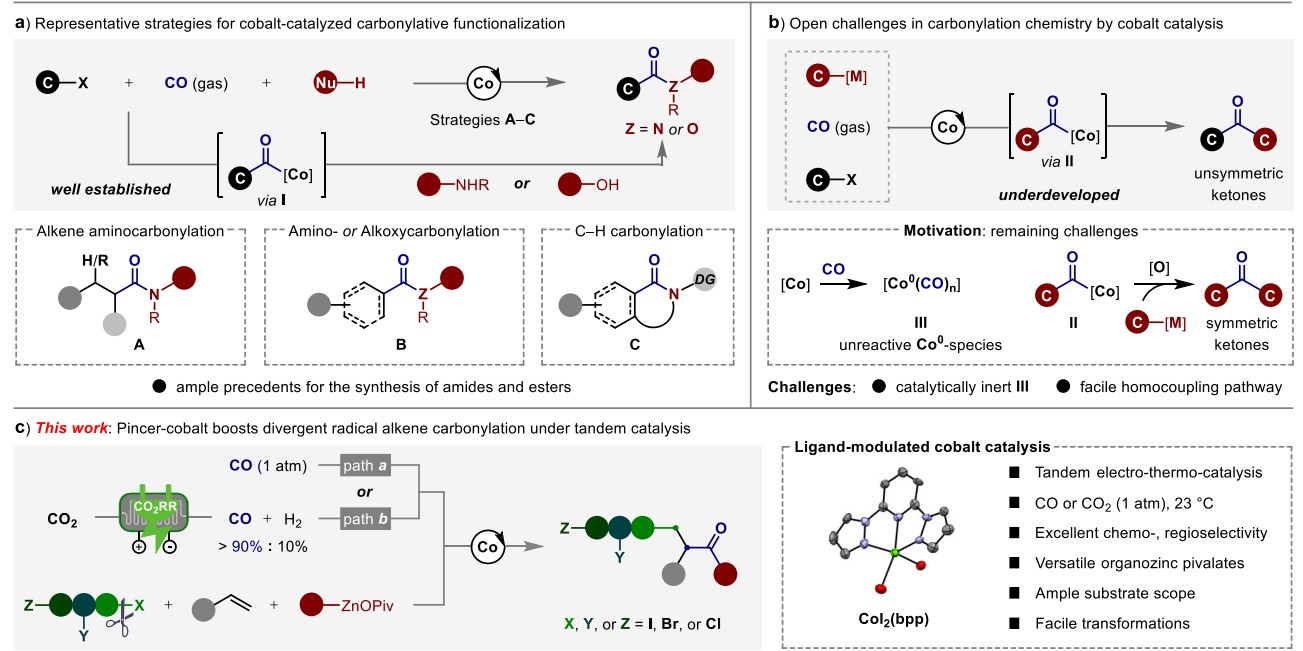

**Fig. 1 | Current advances and limitations of cobalt-catalyzed carbonylative coupling reactions with CO gas. a** Common strategies for cobalt-catalyzed carbonylative functionalization. **b** Open challenges in carbonylation chemistry by cobalt catalysis. **c** This work: pincer-cobalt boosts divergent radical alkene carbonylation under tandem catalysis.

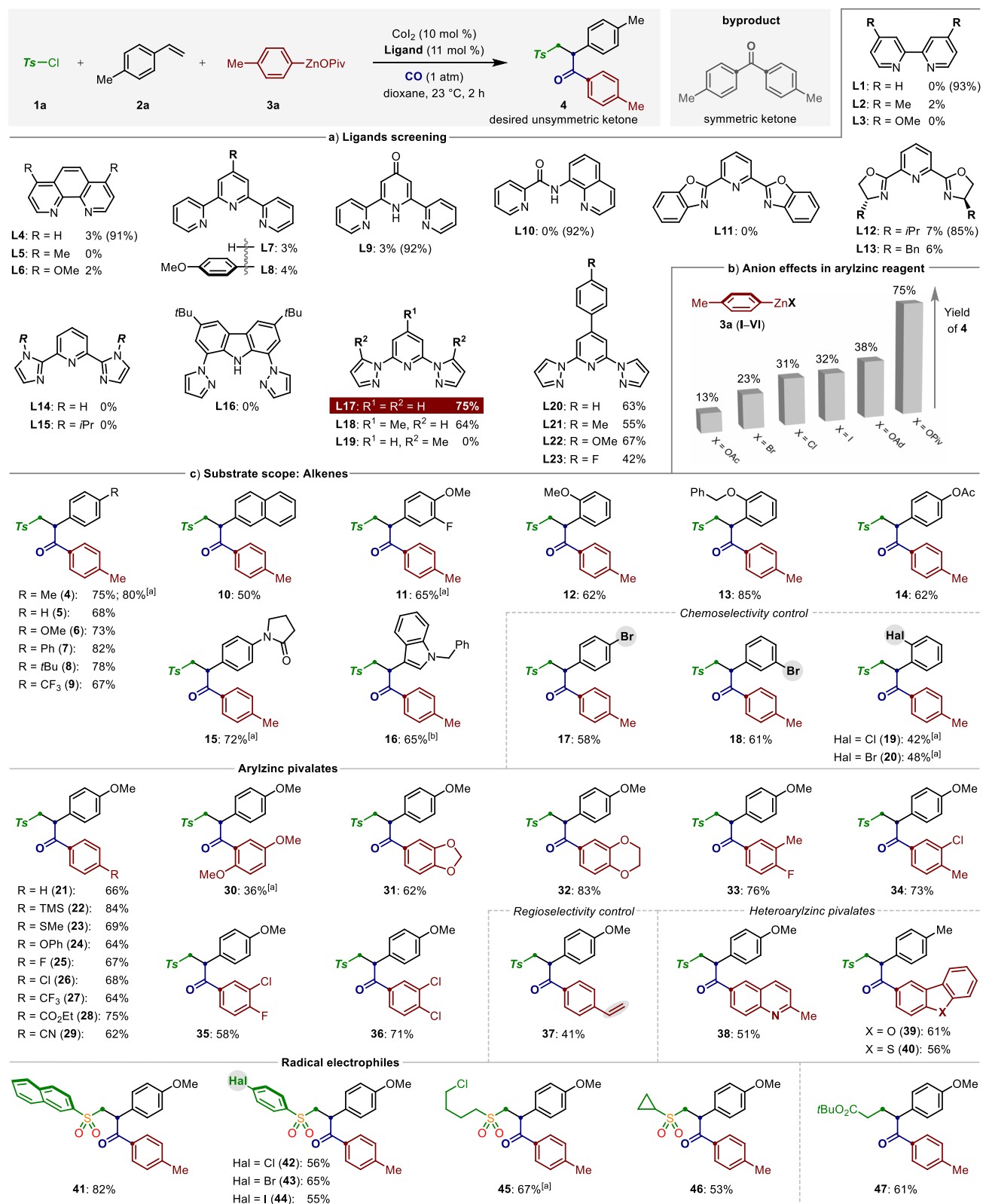

**Fig. 2 | Pincer cobalt-catalyzed alkene carbonylative sulfonylation.**
**a**, **b** Optimization studies. **c** Substrate scope investigation. Reaction conditions: radical electrophile (0.4 mmol, 2.0 equiv), alkene (0.2 mmol, 1.0 equiv), (hetero) aryl–ZnOPiv (0.4 mmol, 2.0 equiv), CoI₂ (10 mol %), bpp (**L17**, 11 mol %), CO (1 atm), 1,4-dioxane, @ 23 °C, 2 h. [a] Ar–ZnOPiv (3.0 equiv). Note: The yields of symmetric ketone are in parentheses.

supported p-tolylzinc reagents, which prepared by transmetalation reactions of p-tolylmagnesium chloride with ZnX₂ (X = Cl, Br, I, OAc, OAd), resulted in significantly decreased yields of the product **4** (Fig. 2b). Generally, the arylzinc reagents prossessing more electron-rich carboxylate anions showed superior reactivity than the halide-supported organozincs. These unique paradigms of anion-effects stand as a treatment to tune the reactivity of organozinc reagents and extend their applications in coupling reactions[58-63].

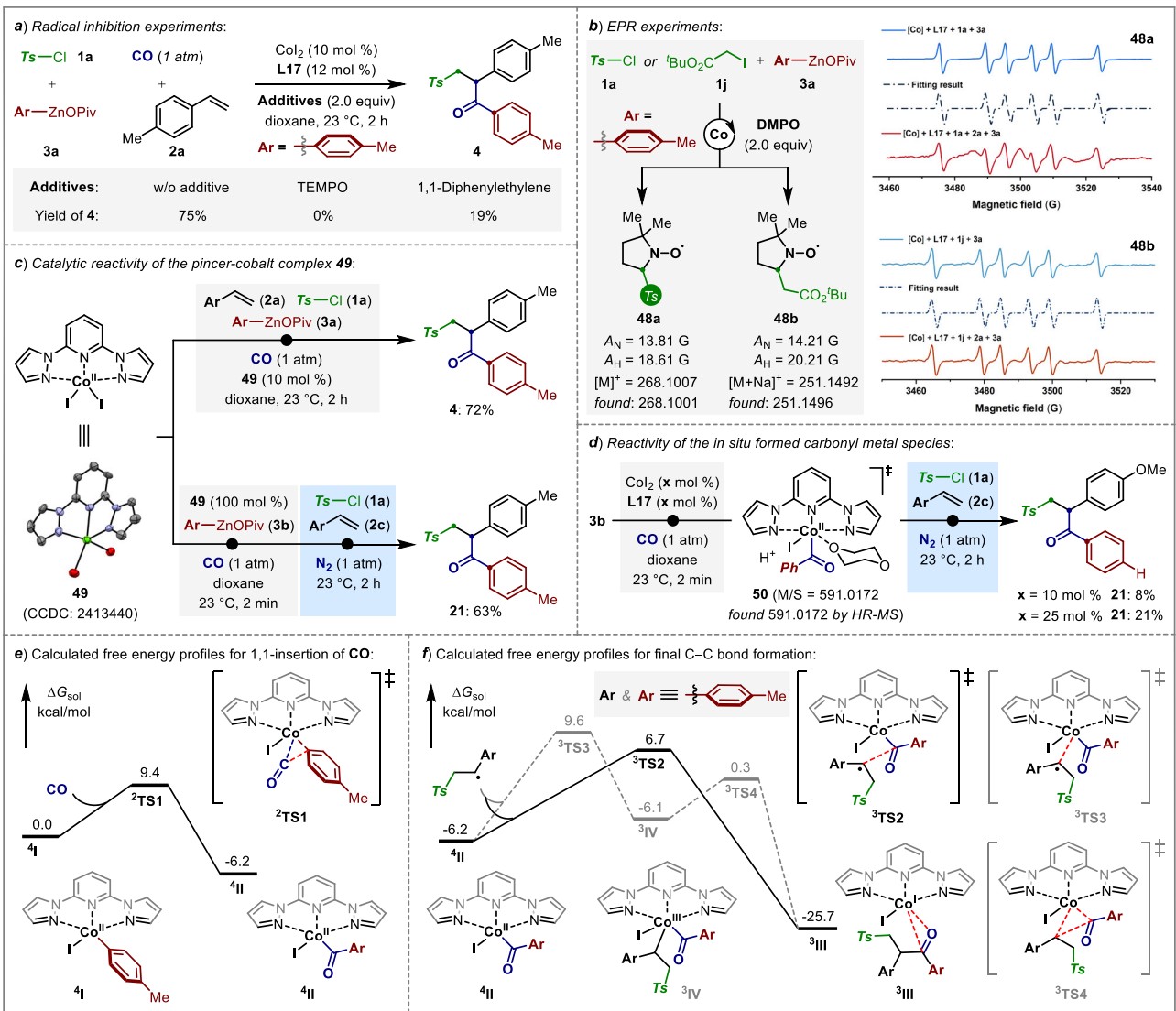

**Fig. 3 | Mechanistic studies. a, b** Radical evidences. **c, d** Control experiments with pincer cobalt complex **49**. **e, f** DFT calculations for the mechanistic studies. The computed energy profiles depict the most favorable spin states of the species, while the energies of the less favorable spin states are detailed in the Supporting Information (see Figs. S27 and S28).

With the optimized pincer-cobalt catalyst in hand, we next turned our attention to its versatility in the cobalt-catalyzed alkene carbonylative sulfonylation reaction. As shown in Fig. 2c, the substrate scope was largely insensitive to the electronic and steric changes on the para-, meta-, and ortho-substituted aryl alkenes (**4–20**). Indeed, vinylarenes bearing acetal (**14**), lactam (**15**), heteroarene (**16**), as well as aryl halides (**17–20**) posed no problems. Notably, no Negishi-type reactions were observed in the latter under the limits of detection. Likewise, the pincer-cobalt catalysis also showed excellent compatibility toward a large chemical space of both electron-rich and electron-poor arylzinc pivalates substituents at the para-, meta-, and ortho-positions (**21–36**). It is noteworthy that a range of valuable functional groups, including silyl moiety (**22**), thiomethyl (**23**), trifluoromethyl (**27**), ester (**28**), nitrile (**29**), dioxole (**31–32**), and halogens (**25**, **26**, **33–36**) were well tolerated under the mild conditions. Moreover, arylzinc pivalate possessing a vinylarene-moiety was identified as viable nucleophile, smoothly underwent highly regioselective carbonylsulfonylation process across the double bond of 4-methylstyrene (**2a**). While the olefin motif derived from arylzinc pivalate side remained untouched, thus affording the ketone **37** as the sole product. Gratifyingly, the quinolone-, dibenzofuran-, and dibenzothiophene-based ketones (**38–40**)

were obtained in 51–56% yields using the corresponding heteroarylzinc pivalates as the nucleophiles. Next, the current carbonylative sulfonylation reaction can be conducted with various sulfonyl chlorides. As shown, sulfonyl chlorides in particularly those bearing aryl halides (**42–44**), alkyl halides (**45**), cyclopropyl (**46**), as well as the primary alkyl iodide (**47**) could be coupled in moderate to high yields. Again, our pincer-cobalt catalyst showed excellent chemoselectivity to differentiate $SO_2$–Cl from other reactive $Csp^2$–Cl, $Csp^2$–Br and $Csp^2$–I bonds, selectively occurring cascade alkene sulfonylcarbonylation.

Intrigued by the efficient catalytic activity of the pincer cobalt catalysis, we sought to unravel the reaction mode of action. To this end, the control experiments with stoichiometric amount of representative radical scavengers were performed (Fig. 3a). Among them, a significantly reduced yields of ketone **4** was observed when employing 1,1-diphenylethene as the additive, while the multicomponent reaction was completely suppressed in the presence of 2,2,6,6-tetramethyl-1-piperinedinyloxy (TEMPO). A further radical-clock experiment with α-cyclopropyl styrene afforded both sulfonylcyclized product and ring-opened sulfonyl-carbonylated product (see SI, Fig. S10). These results strongly consistent with the EPR spin-trapping experiments using tosyl chloride (**1a**) or tert-butyl 2-iodoacetate (**1j**) as the radical precursors.

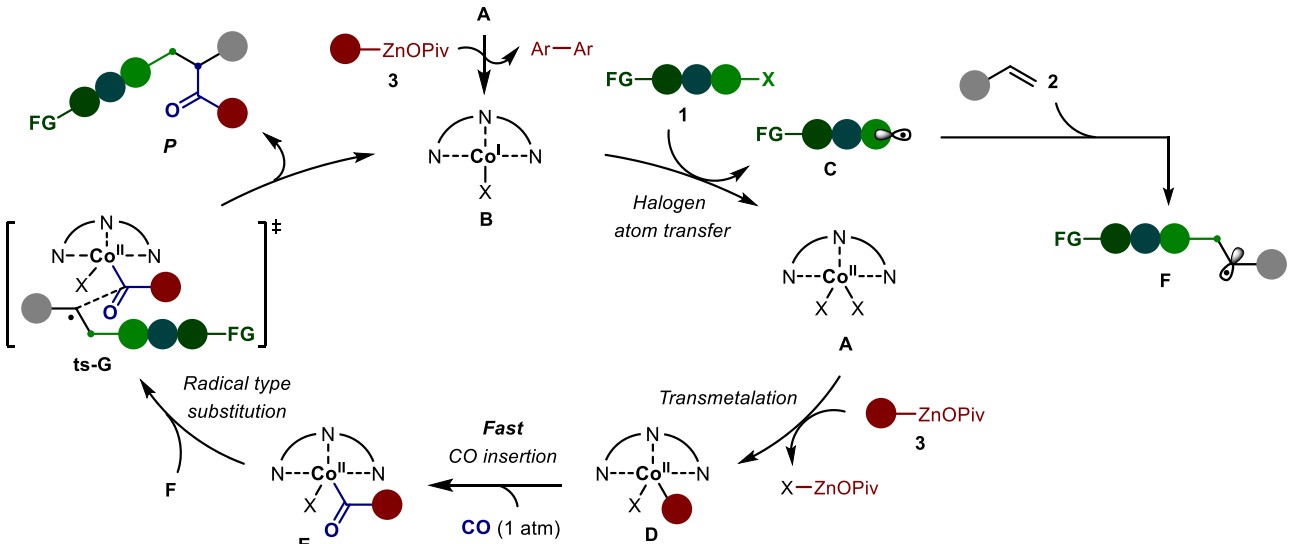

**Fig. 4 | Proposed reaction mechanism.** The in situ formed Co$^I$(bpp) **B** is proposed as the catalytically active catalyst for the halogen atom transfer step to afford the Co$^{II}$(bpp)-species **A**. Sequence transmetalation and CO insertion with **A** generate the key acyl-Co$^{II}$(bpp) intermediate **E**, and a subsequent radical type substitution between **E** and **F** furnishes the product and regenerate intermediate **B**.

Notably, regardless of the absence or presence of alkene, the same sulfonyl- or carbon-centered radical intermediates **48a** ($A_N = 13.81$ G, $A_H = 18.61$ G) and **48b** ($A_N = 14.21$ G, $A_H = 20.21$ G) were captured by 5,5-dimethyl-1-pyrroline-1-oxide (DMPO), thereby confirming the radical's role in this pincer cobalt-catalyzed multicomponent carbonylative functionalization (Fig. 3b)[59,64].

To better elucidate the structure and properties of the pincer-cobalt species relevant to the catalytic reactivity, a dark green crystal of bpp-ligated Co$^{II}$-complex **49** was prepared from CoI$_2$ and bpp with 1:1 ratio. A single-crystal X-ray diffraction study demonstrated that **49** adopts a triclinic geometry,("The structure of **49** was determined by X-ray crystallographic analysis. Deposition Number 2413440 contains the supplementary crystallographic data for this paper. This data is provided free of charge by the joint Cambridge Crystallographic Data Centre and Fachinformationszentrum Karlsruhe Access Structures service www.ccdc.cam.ac.uk/structures.") which showed high catalytic reactivity in the alkene sulfonylcarbonylation reaction, leading to the desired product **4** in 72% yield. Notably, the 1.0 equiv of Co$^{II}$(bpp) **49** initially reacted with phenylzinc pivalate (**3b**) under 1 atm of CO for 2 min, followed by transferring the resulting solution into a divided reaction mixture consisting of tosyl chloride **1a** and alkene **2c** under N$_2$ atmosphere, delivering the desired unsymmetric ketone **21** in 63% yield (Fig. 3c). These findings demonstrated that the relatively more electron-rich property of bpp enables a mild reduction of Co$^{II}$(bpp) by arylzincs to avoid the formation of Co$^0$-species[65], selectively leading to the formation of Co$^I$(bpp) for multicomponent radical relay coupling reaction.

Moreover, in order to understand the mode of pincer-cobalt mediated 1,1-insertion of CO we subsequently devised a step-wise operation (Fig. 3d). A mixture of phenylzinc pivalate and different amounts of CoI$_2$ and bpp under CO atmosphere at 23 °C for 2 min could in situ form the acyl-cobalt species **50**, which was further confirmed by HR-MS analysis. Similarly, the solution was transferred into a divided reaction mixture consisting of tosyl chloride **1a** and alkene **2c** under N$_2$ atmosphere, the ketone **21** (8% or 21%) was obtained in near 1:1 equiv ration to that of CoI$_2$-bpp (10% or 25%), respectively. These findings demonstrated a rapid 1,1-insertion of CO to the formation of acyl-Co$^{II}$ species.

Thereafter, density functional theory (DFT) calculations were conducted to disclose the kinetic model of this process in detail (Fig. 3e). In our calculations, the 1,1-insertion between the quartet

pincer Co$^{II}$-aryl complex $^4$**I** and CO could occur rapidly via transition state $^2$**TS1** to furnish the acyl-Co$^{II}$-species $^4$**II**. The calculated free energy barrier is 9.4 kcal/mol, revealing a favorable process as compared to the radical-type oxidation (20.5 kcal/mol) and radical-type substitution (14.9 kcal/mol) between $^4$**I** and benzylic radical (see SI, Fig. S26). This is the key to suppression of three-component competing reaction without the insertion of CO.

Subsequently, the reaction between the acyl-Co$^{II}$ complex $^4$**II** and the benzylic radical could proceed via either a radical-type oxidation or a radical-type substitution pathway. DFT calculations indicate that the radical-type oxidation involves a high-energy transition state $^3$**TS3** ($\Delta G^\ddagger = 15.8$ kcal/mol) to form the triplet intermediate $^3$**IV**. In contrast, the radical-type substitution of its connected acyl-moiety with the benzylic radical proceeds through the triplet transition state $^3$**TS2** with a lower free energy barrier of 12.9 kcal/mol. These results suggested that the generation of a high valent Co$^{III}$-complex (like $^3$**IV**) and its subsequent reductive elimination pathway could be excluded as the mechanism for the final C – C bond formation (Fig. 3f).

Based on our mechanistic studies, a plausible catalytic cycle for this cobalt-catalyzed four-component carbonylative reaction is proposed in Fig. 4. Initially, due to the suitable stereoelectronic property of pincer ligand and relatively lower reducibility of OPiv-supported arylzincs[57,59], the reduction of Co$^{II}$(bpp) **A** by stioichimetric arylzinc pivalates uniquely dictates the formation of Co$^I$(bpp) **B**, rather than Co$^0$-complex, thereby inhibiting the generation of catalytically inert Co$^0$(CO)$_n$. Subsequently, Co$^I$(bpp) **B** promotes the halogen atom transfer (HAT) with radical precursors (**1**) to afford the radical **C** and releases Co$^{II}$(bpp) **A**. Transmetalation reaction between **A** and arylzinc pivalates furnishes the aryl-Co$^{II}$(bpp) species **D**, which undergoes a fast 1,1-insertion with CO to generate the acyl-Co$^{II}$(bpp) intermediate **E**. Alternatively, automatically radical addition of **C** into alkene (**2**) forms a benzyl radical **F**. A radical-type substitution between **E** and **F** affords the final product via the transition state **G**, and regenerates the Co$^{II}$(bpp) complex **B**.

Encouraged by the high efficacy of pincer-cobalt catalysis in alkene carbonylsulfonylation reaction, we wondered whether it would be possible to directly replace CO with CO$_2$ under tandem electro-thermo-catalysis, which includes CO$_2$ electro-reduction to CO, coupled with cobalt-catalyzed alkene carbonylative functionalization (Fig. 5). Importantly, a commercial available silver powder was tested in a customized flow cell and exhibited satisfactory CO$_2$-to-CO conversion

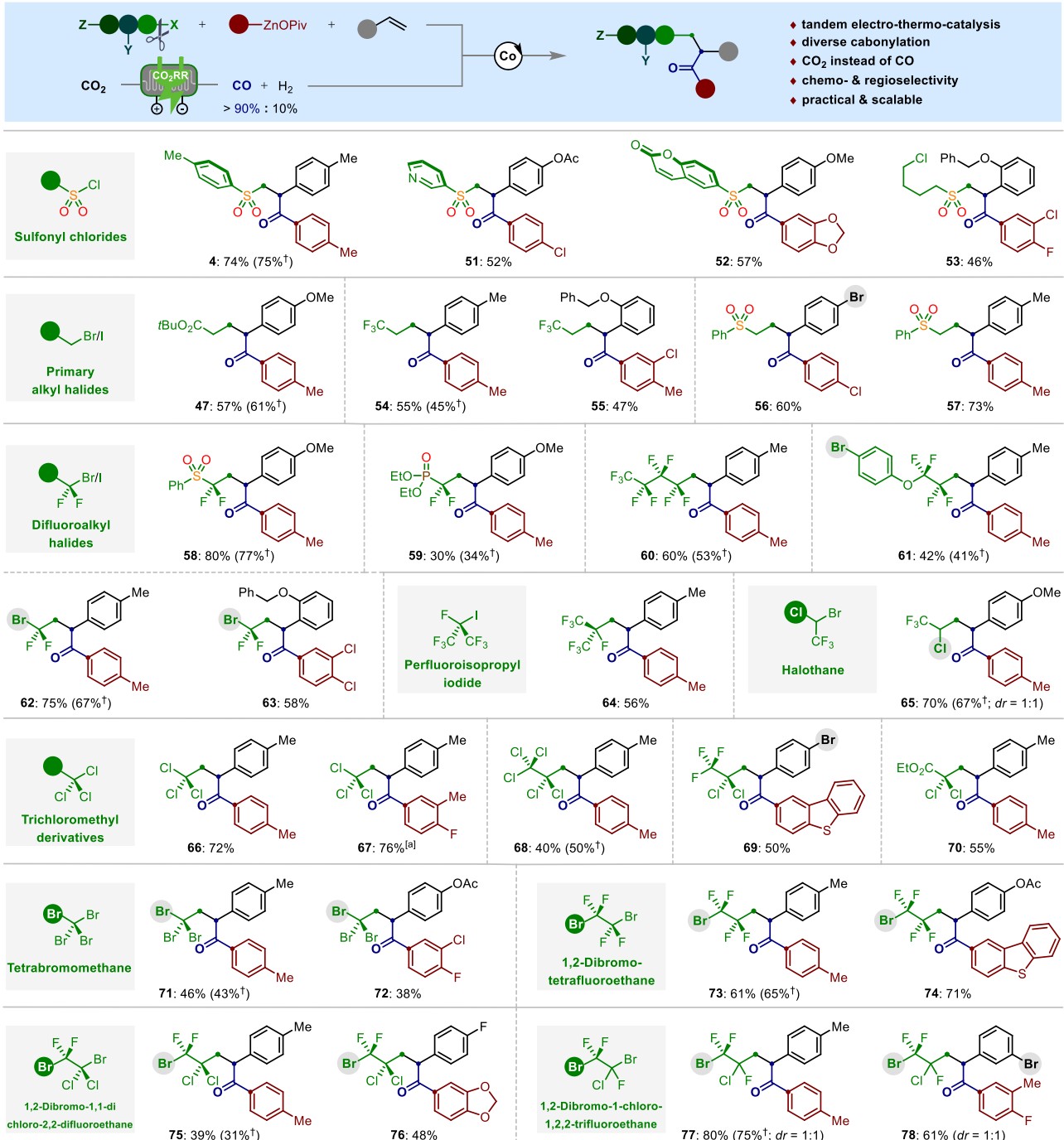

**Fig. 5 | Tandem electro-thermo catalysis enables divergent alkene carbonylative functionalization using CO₂ as the C1 source.** Reaction conditions: Sulfonyl chloride or alkyl halides (0.4 mmol, 2.0 equiv), Ar–ZnOPiv (0.6 mmol, 3.0 equiv), alkene (0.2 mmol, 1.0 equiv), CoI₂ (10 mol %), bpp (**L17**, 12 mol %), CO:H₂ ( > 90:10, 1 atm), 1,4-dioxane, @ 23 °C, 2 h. † Yields in parentheses are obtained under 1 atm of CO gas. [a] 2.0 mmol scale.

at an average rate of 3.73 mmol h⁻¹ for over 40 h, providing a product gas mixture of CO/H₂ with >90/10 ratio. The gas can be collected for tandem reactions after a scrubbing process with concentrated NaOH to remove the unreacted CO₂ carrier gas (see SI, Fig. S17)[66–68]. To our delight, the residual H₂ had no influence for the cascade pincer-cobalt catalysis, thus giving the desired product **4** in 74% yield within equal efficacy to using CO gas. Likewise, other sulfonylcarbonylated compounds **51–53** were easily within reach under the tandem catalysis using CO₂ as the initial C1 synthon. Hence, the upgrading of CO₂ into valuable organic molecules should prove instrumental for potential applications of our tandem electro-thermo catalysis.

It is noteworthy that our tandem catalysis could be extended to achieve alkene carbonylative multicomponent functionalization with a variety of alkyl halides other than sulfonyl chlorides, thus allowing us to rapidly construct complex carbogenic skeletons via insequence C–C bonds formation. As shown, primary alkyl halides accommodating functionalities such as ester (**47**), trifluoromethyl (**54–55**), and sulfone (**56–57**) were proved to be viable substrates, furnishing the desired alkylcarbonylated products in 47–73% yields. Driven by the dramatic effects of organofluorinated compounds in medicinal chemistry[69,70], we investigated the possibility of our protocol for direct access to versatile fluorinated ketones. A wide range of fluoroalkyl

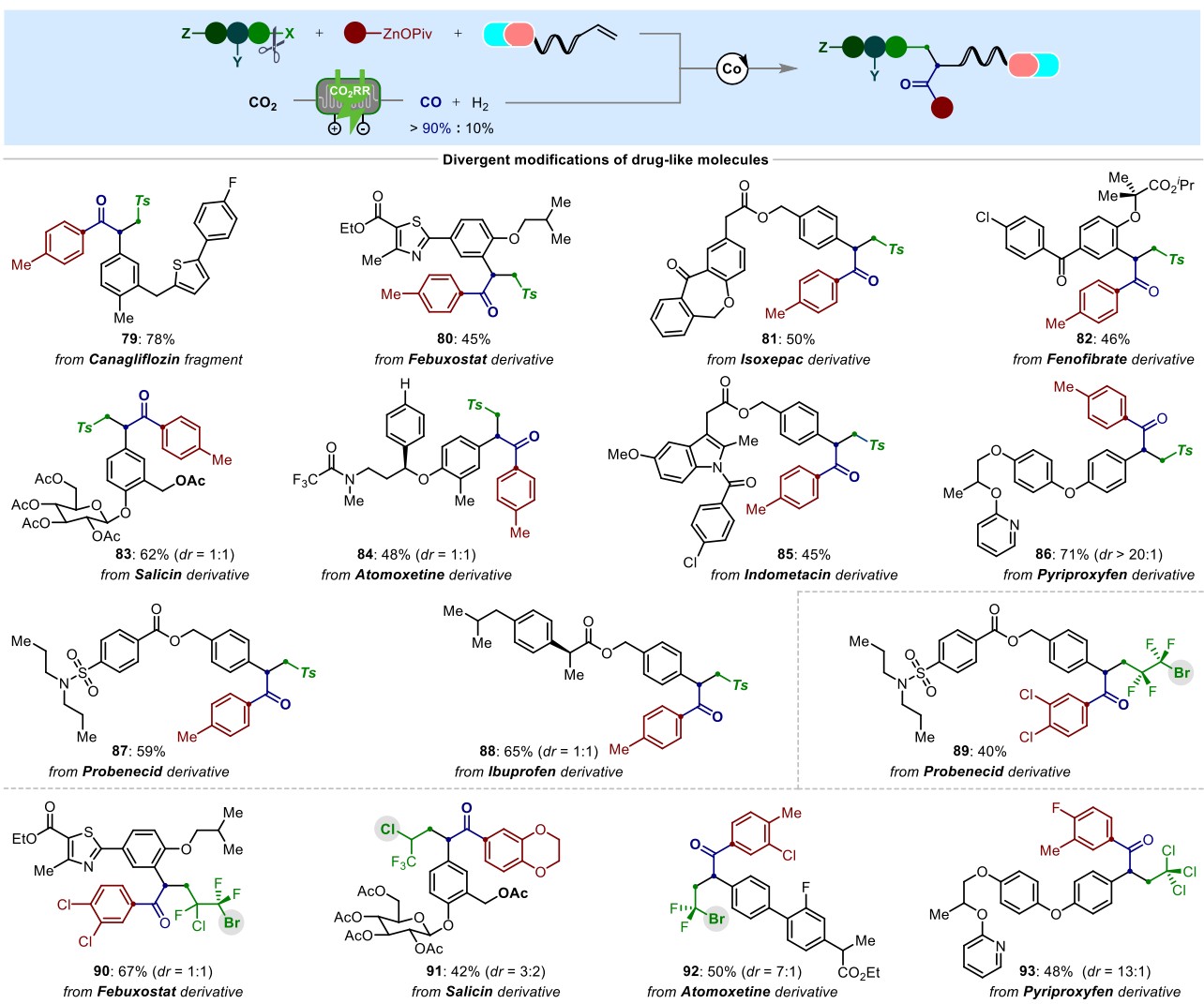

**Fig. 6 | Late-stage diversifications of bioactive molecules.** Reaction conditions: radical precursors (0.4 mmol, 2.0 equiv), Ar–ZnOPiv (0.6 mmol, 3.0 equiv), alkene (0.2 mmol, 1.0 equiv), CoI$_2$ (10 mol %), bpp (**L17**, 12 mol %), CO:H$_2$ ( > 90:10, 1 atm), 1,4-dioxane, @ 23 °C, 2 h.

halides, such as those containing sulfone (**58**), phosphate (**59**), perfluoroalkyl (**60**, **64**), phenoxy (**61**), alkyl bromide (**62** – **63**) and chloride (**65**) proved compatible coupling partners, thereby leading to the desired alkylfluorinated ketones in moderate to good yields. Indeed, the success with dibromodifluoromethane for alkene carbonylative fluoromethylation (**62** – **63**) inspired us whether our protocol could be extended to the regio- and chemoselective transformation with polyhalogenated electrophiles. Intriguingly, trichloromethyl derivatives can also be successfully engaged in this alkene carbonylative alkylation through selective mono Csp³ – Cl bond cleavage fortunately without compromising the catalytic efficiency, led to the alkylchlorinated ketones **66** – **70** in satisfied yields. Furthermore, extensions to those alkylbromides containing multiple reactive Csp³ – Br/Cl bonds were also examined under otherwise identical reaction conditions. Particularly interesting was the unique mono Csp³ – Br bond activation to afford the polyhalo-genated ketones **71** – **78** with complete control of regio-, and chemoselectivity. Notably, the excellent selectivity control of our tandem catalysis toward both Csp³ – X and Csp² – X (X = Br, Cl), as well as wide functional group compatibility leave ample opportunities for increasing functional molecular complexity via late-stage diversifications.

To illustrate the synthetic utility of our tandem electro-thermocatalysis to drug discoveries in medicinal chemistry, the advanced

synthetic alkenes derived from drug-like molecules, such as canagliflozin (**79**), febuxostat (**80**), isoxepac (**81**), fenofibrate (**82**), salicin (**83**), atomoxetine (**84**), indomethacin (**85**), pyriproxyfen (**86**), probenecid (**87**), ibuprofen (**88**), were able to incorporate the sulfonyl and carbonyl fragments across the double bond in moderate to high yields with excellent functional group compatibility. As expected, the more challenging polyhalogenated Csp³-electrophiles were smoothly employed under the catalytic system to afford the polyhalogenated derivatives **89–93** with complete control of regio-, and chemoselectivity. These results should prove the robustness and synthetic application of our protocol towards the construction of highly functionalized molecules in a streamline and diversified fashion, thereby providing a versatile tool for medicinal chemists in their drug-discovery setting (Fig. 6).

To showcase the synthetic utilizations of this cobalt-catalyzed multicomponent carbonylative functionalization reaction in creating high-value carbogenic skeletons, a series of derivatizations with the resulting functionalized ketones were conducted. Firstly, the halogenated ketones could be easily subjected to the intramolecular mono-dehalogenation process, giving the gem-dihalogenated cyclopropanes **94–96** in excellent yields. In particular, the substrates **77** and **69** possessing multiple reactive Csp³ – X (X = Br or Cl) bonds, selectively underwent mono Csp³ – Cl bond cleavage to afford the halogenated

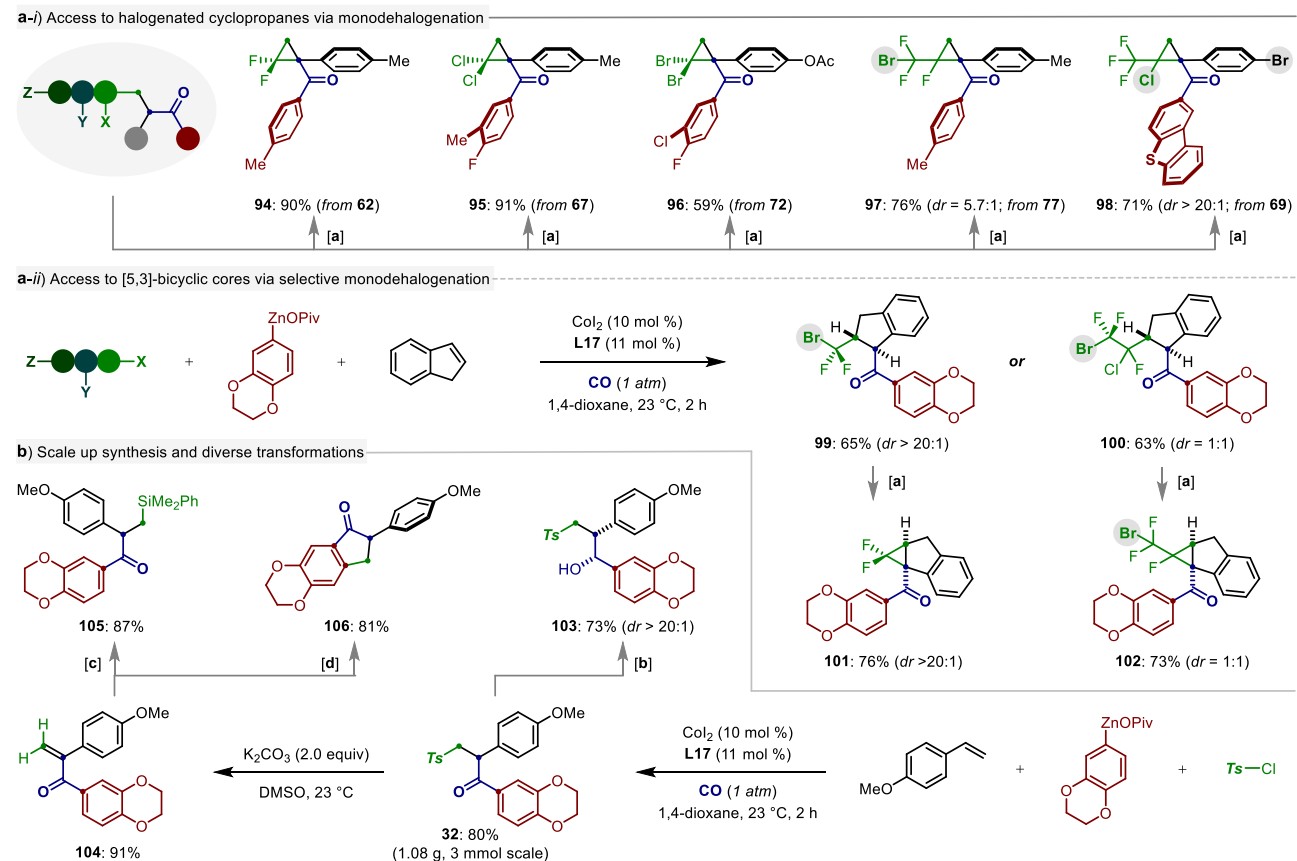

**Fig. 7 | Diverse synthetic utilizations. a** Access to halogenated cyclopropanes via monodehalogenation. **b** Scale up synthesis and diverse transformations. Reaction conditions: [a] Cs₂CO₃ (3.0 equiv), DMSO, @ 25 or 65 °C, 12 h. [b] NaBH₄ (2.0 equiv), MeOH, @ 23 °C, 4 h. **c** Me₂PhSi−ZnOPiv (1.2 equiv), CuI (10 mol %), THF, @ 23 °C, 4 h. **d** con. H₂SO₄, DCM-EtOH, @ 0 – 23 °C, 1 h.

cyclopropanes **97–98** as the sole products with moderate to high diastereoselectivity (Fig. 7a–i). It is worth noting that various poly-halogenated alkanes, along with arylzinc pivalate under 1 atm of CO gas, are capable of the alkylative carbonylation of indene and delivering **99–100** in good yields and diastereoselectivities. Follow-up monodehalogenation process provided an expedient route to the[3,5]-bicyclic ketones **101–102** in 73–76% yields (Fig. 7a-ii). Importantly, this multicomponent transformation can be easily scaled up to gram scale, yielding the product **32** with equal efficiency. The newly formed ketone functionality of **32** could be reduced in the presence of NaBH₄ to afford alcohol **103** in excellent diastereoselectivity (dr >20:1). Moreover, facile desulfonylation process allowed for synthesis of α,β-unsaturated ketone **104**, which could be further modified by copper-catalyzed silylation or Friedel-Crafts reaction to produce the products **105** and **106** (Fig. 7b), respectively.

## Discussion

In conclusion, we have demonstrated that an industrial-friendly cobalt catalysis for the divergent carbonylative functionalization of alkenes under 1 atm of CO gas. Herein, the use of tridentate NNN-type pincer ligand is the key to tune the catalytic reactivity of cobalt for selectively dictating a full multi-component radical relay carbonylation reaction. Importantly, direct use CO₂ as the C1 source in the multicomponent carbonylation reaction can be achieved under a tandem electro-thermo-catalysis, thus allowing us to rapidly and reliably construct unsymmetric ketones with ample substrate scope of alkenes, organozinc reagents, as well as sulfonyl and alkyl radical precursors, particularly in a regio- and chemoselective fashion. Moreover, the synthetic utility of this approach was well illustrated by the late-stage

modifications of bioactive molecules and the facile transformations of the resulting ketones.

## Methods

### General procedure for cobalt-catalyzed alkene carbonylation under 1 atm of CO

An oven-dried tube was charged with CoI₂ (10 mol %), **L17** (12 mol %), sulfuryl chloride 1 (0.4 mmol, 2.0 equiv). Then the tube was evacuated and backfilled with CO (three times, 1 atm, balloon). Anhydrous 1,4-dioxane (1.0 mL) was added and stirred for 10 minutes vigorously. Alkene 2 (0.2 mmol, 1.0 equiv) was added. Then arylzinc pivalates 3 (0.4 mmol, 2.0 equiv) resolved in 1,4-dioxane (0.5 mL) was added dropwise over 5 minutes. The reaction mixture was stirred at 23 °C for 2 h. When the reaction was completed, the resulting residue was purified by column chromatography on silica gel (petroleum ether/EtOAc) to yield products.

### General procedure for cobalt-catalyzed alkene carbonylation Using CO₂ as the C1 source

An oven-dried tube was charged with CoI₂ (10 mol %), L17 (12 mol %). Then the tube was evacuated and backfilled with the gas mixture of CO:H₂ (>90:10 ratio, three times, 1 atm, balloon). Anhydrous 1,4-dioxane (1.0 mL) was added and stirred for 10 minutes vigorously. Radical source 1 (0.4 mmol, 2.0 equiv) and alkene 2 (0.2 mmol, 1.0 equiv) were addded. Then arylzinc pivalates 3 (0.6 mmol, 3.0 equiv) resolved in 1,4-dioxane (0.5 mL) was added dropwise over 5 minutes. The reaction mixture was stirred at 23 °C for 2 h. When the reaction was completed, the resulting residue was purified by column chromatography on silica gel (petroleum ether/EtOAc) to yield products.

## Data availability

The authors declare that all the data supporting the findings of this study, including experimental procedures and compound characterization are available within the article and the Supplementary Information. Source data are provided with this pape. The X-ray crystallographic data for structure 49 used in this study are available in the joint Cambridge Crystallographic Data Centre (CCDC 2413440) and Fachinformationszentrum Karlsruhe Access Structures service (www.ccdc.cam. ac.uk/structures). All data are available from the corresponding author upon request. Source data are provided with this paper.

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

## Acknowledgements

We thank the National Natural Science Foundation of China (22322108 to J.L.), Natural Science Foundation of Jiangsu Province (BK20231521 and BK20221355 to J.L.) and Science and Technology Program of Suzhou (ZXL2024399 to J.L.) for financial supports. We also thank Hefei Advanced Computing Center for computational support.

## Author contributions

J.L. conceived and directed the project and wrote the manuscript by all the other authors; S.G., Z.C., and L.P. developed and performed the catalytic methods and the synthetic applications. S.G., K.C., Z.C., and S.D. performed the mechanistic studies; Y.H. and L.H. designed the CO₂-to-CO conversion methods; G.L. designed and directed the DFT calculations; X.W. performed the DFT calculations; S.G. and Z.C. prepared the starting materials, C.N. analyzed the crystal structures, all the authors were involved in interpretation of the results presented in the manuscript.

## Competing interests

The authors declare no competing interests.

## Additional information

https://doi.org/10.1038/s41467-025-63875-4).

