## [Transparent Peer Review file · Nature Communications]

Pincer-Cobalt Boosts Divergent Alkene Carbonylation under Tandem Electro-Thermo-Catalysis

Corresponding Author: Professor Jie Li

Version 0:

Reviewer comments:

Reviewer #1

(Remarks to the Author)

The manuscript by Li, He, and coworkers presents an efficient approach to synthesizing unsymmetric ketones via cobalt-catalyzed four-component carbonylation, using CO as the carbonyl source. This work represents a significant advancement in the field of multicomponent carbonylation reactions. While noble metals are commonly employed as catalysts in such reactions, the development of base metal catalysts, as demonstrated here, offers a more sustainable and cost-effective alternative. The authors' use of a tridentate NNN-type pincer ligand effectively prevents the formation of inert $\text{Co}(\text{CO})_n$ species, a common challenge in cobalt-catalyzed reactions. The catalytic system enables the assembly of styrenes, arylzinc reagents, and electrophiles (e.g., sulfonyl chlorides, (fluoro)alkyl halides) under mild conditions (1 atm CO). The protocol exhibits broad substrate scope and high functional group tolerance, including complex molecules, thus complementing the pioneering work of Zhang using a nickel catalyst. Additionally, the authors demonstrate the versatility of their method by using CO_2 as the carbonyl source under tandem electro-thermo-catalysis. The synthetic utility of the resulting carbonyl compounds is further highlighted by their diverse transformations, such as the generation of halogenated cyclopropanes. Extensive mechanistic studies provide valuable insights into the reaction mechanism.

Overall, this manuscript represents a significant contribution to the field of cobalt-catalyzed multicomponent carbonylation reactions and is recommended for publication in Nat. Commun. after addressing the following revisions:

1. The study focuses on styrenes as substrates. It would be beneficial to explore the applicability of aliphatic alkenes to enhance the impact of this approach.
2. The authors propose that the Co(I) complex B initiates the reaction through an XAT pathway with the electrophiles. However, an alternative pathway beginning with transmetallation followed by XAT is also plausible. Additionally, the possibility of reaction initiation from Co(II) should be considered. In this scenario, the reaction begins with the formation of intermediate E, which reacts with the electrophile to generate a radical and Co(III). This pathway is similar to the nickel-catalyzed pathway reported by Zhang (ref 51).

Reviewer #2

(Remarks to the Author)

The paper titled "Pincer-Cobalt Boosts Divergent Alkene Carbonylation under Tandem Electro-Thermo-Catalysis" by Ge and coworkers describes the radical carbonylation of alkenes catalyzed by cobalt-based catalysts. Through a wide ligand screening, the authors found that the tridentate 2,6-bis(N-pyrazolyl)pyridine (bpp) ligand successfully drives the multicomponent carbonylative coupling reaction to afford the desired unsymmetrical ketone under mild reaction conditions (1 atm CO and RT). The authors performed a thorough substrate scope concerning the alkene, arylzinc, and radical electrophile, demonstrating that the pincer-cobalt catalysis shows excellent compatibility toward a large chemical space of both electron-rich and electron-poor reactants. The authors also demonstrated the compatibility of the developed method with the $\text{CO}:\text{H}_2$ mixture resulting from the electroreduction of CO_2 using a flow cell and a silver-based cathode. Such a tandem approach could be extended to achieve alkene carbonylative functionalization with various alkyl halides other than sulfonyl chlorides as radical electrophiles, thus allowing for rapid construction of complex carbogenic skeletons. The authors went on to demonstrate the applicability of this multicomponent carbonylation reaction to the late-stage functionalization of bio-active molecules by incorporating the sulfonyl and carbonyl fragments across the double bond, obtaining moderate to high yields with excellent functional group compatibility. Finally, the authors also demonstrated the synthetic utilizations of this cobalt-catalyzed multicomponent carbonylative functionalization by a series of derivatizations with the resulting

functionalized ketones. In summary, this work establishes an unprecedented cobalt-based catalytic method for the divergent carbonylative functionalization of alkenes with a broad substrate scope of alkenes, organozinc reagents, as well as sulfonyl and alkyl radical precursors, for the synthesis of unsymmetric ketones. However, before its publication in Nat. Commun., the authors should address the following comments regarding the results from the mechanistic investigation:

1. The authors claim that the main role of L17 (bpp) in allowing the desired carbonylative reaction is to prevent the over-reduction of the Co(II) complex toward the formation of carbonyl Co(0) species. In Scheme 2, the authors show all the tested ligands that did not yield the desired unsymmetrical ketone (L1-L16). Some of these ligands have very similar stereoelectronic properties to L17. Therefore, the authors should measure the standard potential of representative examples of Co complexes containing such unsuccessful ligands in order to corroborate that the resulting Co catalyst with L17 displays a more negative standard potential than the inactive complexes.
2. In Scheme 2b, the authors show the catalytic results for screening the anion of the p-tolylzinc reagent. On page 3, lines 86-89, the authors only describe the obtained results. The authors should expand this discussion and analyze the obtained trend regarding the electronic effects of the anion featured in the organozinc reagent.
3. The author should discuss the EPR spin-trapping experiments in more detail, particularly the EPR spectra shown in Scheme 3b. The authors did not discuss the EPR spectra recorded in the presence of the alkene, nor the fitted EPR spectra. Additionally, the author should clarify why only MS data is reported for 48a and not for 48b.
4. In the proposed reaction mechanism displayed in Scheme 4, the authors claim that the CoI intermediate generates the radical species, thus yielding a CoII intermediate, which carries out the transmetalation and CO insertion reactions. However, the stoichiometric experiments in Scheme 3d and the bottom part of Scheme 3c show that a CoII species can generate the corresponding radical. Therefore, it is possible that the initial CoII complex (A in Scheme 4) is able to initiate the catalytic cycle, followed by the reduction of the resulting CoIII to CoII complex by the arylzinc reagent. To confirm this alternative pathway, the author should carry out the EPR spin-trapping experiments in the absence of the arylzinc reactant to investigate if the first reduction step (CoII to CoI) is needed to start the catalytic cycle.
5. In the stoichiometric experiments with the isolated Co complex 49, the author should indicate why, in the sequential reaction, the organozinc substrate 3b was used instead of 3a, as in the non-sequential case.
6. On page 6, lines 161-163, the authors state: "Initially, due to the lower reducibility of OPiv-supported arylzincs, the reduction of CoII(bpp) A by stoichiometric arylzinc pivalates uniquely dictates the formation of CoI(bpp) B, rather than Co0-complex, thereby inhibiting the generation of catalytically inert Co0(CO)_n." This sentence contradicts the general idea of the manuscript, which indicates that the relatively more electron-rich property of the bpp ligand prevents an overreduction to Co0 species. The authors should address this inconsistency.
7. The sentence on page 8, lines 182-183, needs to be revised since it does not fit within the paragraph's discussion.
8. The authors should define the DMPO abbreviation.
9. In Scheme 3b, for the generation of molecule 48b, the radical electrophile 1j was employed. The structure or name of 1j is not mentioned anywhere in the text.
10. Several spelling mistakes in the main text, including in the title of the manuscript, must be revised.

Reviewer #3

(Remarks to the Author)

Li, He, Luo, and co-workers demonstrated significant advancements in cobalt-catalyzed multicomponent carbonylation chemistry, demonstrating a practical and efficient method for divergent alkene carbonylative functionalization under mild conditions. Either CO or CO₂ could be utilized as C1 source in this multicomponent alkene carbonylation. With careful modulation of ligand, the toxicity of CO towards metal and side reaction generating symmetry ketones could be avoided. Mechanistic studies through the combinations of EPR, radical clock experiments, HR-MS, and DFT calculations clarify the radical nature of the reaction mechanism and provide insights into the sequence of elementary steps. I would like to strongly recommend its publications on Nature Communications with the following suggestions:

1. The authors reported their DFT calculations on other spin states for radical-type oxidation and radical-type substitution steps in the Supplementary Information. The preference of high spin states is suggested to be discussed in the manuscript and be referred to corresponding figures in SI.
2. The catalytic species, cobalt complex here, should be depicted on-cycle rather than off-cycle. I suggested the authors move the evolution of cobalt complex to the catalytic cycle and redraw the radical relay process as a branched pathway.
3. It is recommended to denote the transition state with a double dagger symbol written as a superscript instead of normal size when discussing energy barrier.

Reviewer #4

(Remarks to the Author)

In this manuscript, the authors report a cobalt-catalyzed alkene carbonylation achieved through the reaction of alkenes with p-toluenesulfonyl chloride (TsCl) and aryl zinc reagents. Notably, other halide sources capable of initiating hydrogen atom transfer to generate radicals could also be applicable. The reaction proceeds in the presence of CoI₂ and an NNN-type ligand, with 2,6-pyrazolopyridine identified as the most effective among the ligands screened. Additionally, an investigation into the anion effect of the organozinc reagents indicates that arylzinc pivalate is the optimal choice. While related alkene carbonylation reactions using Nickel/NNN ligands have been previously reported (ACS Catal. 2023, 13, 4111–4119, J. Am. Chem. Soc. 2020, 142, 18191–18199), the innovative use of CO₂ as a CO surrogate, supported by theoretical insights and the successful late-stage functionalization of bioactive molecules, enhances the significance of this work. Therefore, the manuscript could be suitable for publication in Nature Communications, provided the following revisions are addressed.

1. Is the methodology applicable to vinyl or alkyl zinc compounds instead of aryl zinc?
2. Is the reaction compatible with aliphatic alkenes beyond styrene analogues?
3. The author proposed "CoII(bpp) A by stoichiometric arylzinc pivalates uniquely dictates the formation of CoI(bpp) B"----A is probably the complex 49. What happens if this complex 49 separately treated with arylzinc species? Did the author able to detect CoI(bpp). The study of the mechanism of this step and detection/isolation of Co(I) species is suggested. For this particular step is there any radical inhibition.
4. The other cobalt precursor such as CoBr₂ was found completely inactive. What is the possible reason? Whether from A/49 to the formation of E getting difficult or A to B or halogen transfer step is getting difficult. Experimental or theoretical investigations to clarify this aspect are recommended.
5. For the formation of complex 49 from CoI₂ and the ligand refluxing in the CH₃CN is required. For Scheme 3 experiment d, is in only 2 m at 23 °C. what happen if the reaction time is longer.
6. It is surprising that the authors only screened NNN ligands, especially since the yields for several compounds (e.g., 10, 19, 20, 30) are moderate. Why were PNP cobalt complexes not explored as potential catalysts for this transformation?
7. A dioxane-coordinated complex was detected by HRMS. It may be more appropriate to base the theoretical calculations on this complex; however, further input from a theoretical expert could help determine the most suitable approach.
8. In the theoretical pathway from 4II to 3IV, is there a possibility of forming a Co(II) intermediate through the dissociation of species I? This alternative route involving Co(II), rather than Co(III), might represent a lower-energy pathway.

Version 1:

Reviewer comments:

Reviewer #1

(Remarks to the Author)

The authors have addressed the issues raised by the referees. The manuscript is now suitable for publication.

Reviewer #2

(Remarks to the Author)

The authors have addressed all of my comments. I recommend this work for publication in Nature Communications.

Reviewer #3

(Remarks to the Author)

The authors have fully addressed the reviewers' comments in the revised version of the manuscript. Therefore, I have no further comments.

Reviewer #4

(Remarks to the Author)

The author addresses all the points raised by this reviewer. I recommend its acceptance in Nature Communications in the present form.

| Ren-Ai Road 199 | 215123 Suzhou

College of Chemistry, Chemical Engineering and
Materials Science, Soochow University

Prof. Dr. Jie Li

Ren-Ai Road 199
215123 Suzhou, P. R. CHINA

Tel.: +86 (0) 512 65880089

Fax: +86 (0) 512 65880089

E-mail: jjackli@suda.edu.cn

Internet: <https://chemistry.suda.edu.cn>

Point-by-Point Responses

Reviewer #1 (Remarks to the Author):

Comments: The manuscript by Li, He, and coworkers presents an efficient approach to synthesizing unsymmetric ketones via cobalt-catalyzed four-component carbonylation, using CO as the carbonyl source. This work represents a significant advancement in the field of multicomponent carbonylation reactions. While noble metals are commonly employed as catalysts in such reactions, the development of base metal catalysts, as demonstrated here, offers a more sustainable and cost-effective alternative. The authors' use of a tridentate NNN-type pincer ligand effectively prevents the formation of inert $\text{Co}(\text{CO})_n$ species, a common challenge in cobalt-catalyzed reactions. The catalytic system enables the assembly of styrenes, arylzinc reagents, and electrophiles (e.g., sulfonyl chlorides, (fluoro)alkyl halides) under mild conditions (1 atm CO). The protocol exhibits broad substrate scope and high functional group tolerance, including complex molecules, thus complementing the pioneering work of Zhang using a nickel catalyst. Additionally, the authors demonstrate the versatility of their method by using CO_2 as the carbonyl source under tandem electro-thermo-catalysis. The synthetic utility of the resulting carbonyl compounds is further highlighted by their diverse transformations, such as the generation of halogenated cyclopropanes. Extensive mechanistic studies provide valuable insights into the reaction mechanism. Overall, this manuscript represents a significant contribution to the field of cobalt-catalyzed multicomponent carbonylation reactions and is recommended for publication in *Nat. Commun.* after addressing the following revisions.

Response: Many thanks for this referee's positive comments.

Q1) The study focuses on styrenes as substrates. It would be beneficial to explore the applicability of aliphatic alkenes to enhance the impact of this approach.

Response: Thanks for the comments. As shown in Scheme S1, aliphatic alkene, as well as various alkenes bearing a heteroatom-containing directing group have been utilized as the unsaturated substrates under the standard reaction conditions. However, no positive results for the envisioned cobalt-catalyzed four-component carbonylative functionalization reaction were detected. The corresponding alkenes remained untouched in the reaction mixture, which can be also detected by GC-analysis, only forming the symmetric diarylmethyl ketone as the byproduct.

Scheme S1. Unsuccessful substrates of olefins.

Moreover, ethylene was also used as the unsaturated substrate for the envisioned four-component carbonylative transformation. Unfortunately, no desired product was observed under 1 atm or 5 atm of ethylene and CO gaseous mixture (Scheme S2). We have added these limitations in the revised Supporting Information.

Scheme S2. Multicomponent carbonylation with ethylene as the unsaturated substrate.

Q2) The authors propose that the Co(I) complex **B** initiates the reaction through an XAT pathway with the electrophiles. However, an alternative pathway beginning with transmetalation followed by XAT is also plausible. Additionally, the possibility of reaction initiation from Co(II) should be considered. In this scenario, the reaction begins with the formation of intermediate **E**, which reacts with the electrophile to generate a radical and Co(III). This pathway is similar to the nickel-catalyzed pathway reported by Zhang (ref 51).

Response: Thanks for the suggestions.

Firstly, we have performed a series of EPR spin-trapping experiments using tosyl chloride (**1a**) or *tert*-butyl 2-iodoacetate (**1j**) as the radical precursors under pincer-cobalt catalysis. Notably, both sulfonyl- or carbon-centered radical intermediates **48a** and **48b** could be observed in the presence of arylzinc pivalate **3a**. In sharp contrast, no radical signals were detected in the absence of arylzinc pivalate **3a** (Scheme S3). These observations should prove that the Co(II)-complex is not the suitable metal-source for triggering the generation of radicals. Therefore, the Co(II/III) catalytic cycle can be excluded.

In addition, as for the initial alternative transmetalation step to form the aryl-Co(I) species or acyl-Co(I) species, we prefer to believe that these species might be the intermediates to promote the oxidative cross-coupling side process, thereby furnishing the symmetric ketones.

Scheme S3. EPR experiments. Note: we have added these results in the revised Supporting Information.

Reviewer #2 (Remarks to the Author):

Comments: The paper titled “Pincer-Cobalt Boosts Divergent Alkene Carbonylation under Tandem Electro-Thermo-Catalysis” by Ge and coworkers describes the radical carbonylation of alkenes catalyzed by cobalt-based catalysts. Through a wide ligand screening, the authors found that the tridentate 2,6-bis(*N*-pyrazolyl)pyridine (bpp) ligand successfully drives the multicomponent carbonylative coupling reaction to afford the desired unsymmetrical ketone under mild reaction conditions (1 atm CO and RT). The authors performed a thorough substrate scope concerning the alkene, arylzinc, and radical electrophile, demonstrating that the pincer-cobalt catalysis shows excellent compatibility toward a large chemical space of both electron-rich and electron-poor reactants. The authors also demonstrated the compatibility of the developed method with the CO:H₂ mixture resulting from the electroreduction of CO₂ using a flow cell and a silver-based cathode. Such a tandem approach could be extended to achieve alkene carbonylative functionalization with various alkyl halides other than sulfonyl chlorides as radical electrophiles, thus allowing for rapid construction of complex carbogenic skeletons. The authors went on to demonstrate the applicability of this multicomponent carbonylation reaction to the late-stage functionalization of bioactive molecules by incorporating the sulfonyl and carbonyl fragments across the double bond, obtaining moderate to high yields with excellent functional

group compatibility. Finally, the authors also demonstrated the synthetic utilizations of this cobalt-catalyzed multicomponent carbonylative functionalization by a series of derivatizations with the resulting functionalized ketones. In summary, this work establishes an unprecedented cobalt-based catalytic method for the divergent carbonylative functionalization of alkenes with a broad substrate scope of alkenes, organozinc reagents, as well as sulfonyl and alkyl radical precursors, for the synthesis of unsymmetric ketones. However, before its publication in *Nat. Commun.*, the authors should address the following comments regarding the results from the mechanistic investigation

Response: We thank this reviewer for the kind comments.

Q1) The authors claim that the main role of **L17** (bpp) in allowing the desired carbonylative reaction is to prevent the over-reduction of the Co(II) complex toward the formation of carbonyl Co(0) species. In Scheme 2, the authors show all the tested ligands that did not yield the desired unsymmetrical ketone (**L1–L16**). Some of these ligands have very similar stereoelectronic properties to **L17**. Therefore, the authors should measure the standard potential of representative examples of Co complexes containing such unsuccessful ligands in order to corroborate that the resulting Co catalyst with **L17** displays a more negative standard potential than the inactive complexes.

Response: Thanks for the comments and suggestions.

As shown in Scheme S4, cyclic voltammetry (CV) experiments were performed to explore the electronic properties of Co^{II}-complexes with different *N,N,N*-type ligands (**L7**, **L11**, **L15** or **L17**). Preliminary studies demonstrated that the reductive peak of Co^{II}(**L7**) to Co^I(**L7**) was increased to -1.14 V, while the reductive peaks in the presence of these more electron-rich *N,N,N*-type ligands **L11**, **L15** and **L17** were decreased to -2.26, -2.14 and -2.03 V, respectively. These results might suggest that the Co^{II}(**L7**) is easier to be reduced to the catalytically inert Co⁰-species. Compared to the Co^{II}(**L11**) and Co^{II}(**L15**), Co^{II}(**L17**) is relatively easier to be reduced to the catalytically active Co^I(**L17**)-species. Indeed, the paradigms of ligand-modulation might be very complicated, our findings are very preliminary clues to illustrate the effect of ligand-modulation in this cobalt-catalyzed multicomponent carbonylative transformation.

Scheme S4. Cyclic voltammetry studies with different Co(II)-complexes.

Q2) In Scheme 2b, the authors show the catalytic results for screening the anion of the *p*-tolylzinc reagent. On page 3, lines 86-89, the authors only describe the obtained results. The authors should expand this discussion and analyze the obtained trend regarding the electronic effects of the anion featured in the organozinc reagent.

Response: Thanks for the suggestions. As shown in Scheme 2b, the desired four-component coupling product **4** was obtained in 38–75% yields when employing OAd- or OPiv-supported arylzinc reagents as the nucleophilic partners. In contrast, the halide-supported *p*-tolylzinc reagents only afforded the

desired product **4** in 23–32% yields. These results might suggest that the more electron-donating carboxylate anions superior than the halide anions in tuning the reactivity of arylzinc reagents. As such, we have added the following description in the revised manuscript.

“Generally, the arylzinc reagents possessing more electron-rich carboxylate anions showed superior reactivity than the halide-supported organozincs. These unique paradigms of anion-effects stand as a treatment to tune the reactivity of organozinc reagents and extend their applications in coupling reactions.”.

Q3) The author should discuss the EPR spin-trapping experiments in more detail, particularly the EPR spectra shown in Scheme 3b. The authors did not discuss the EPR spectra recorded in the presence of the alkene, nor the fitted EPR spectra. Additionally, the author should clarify why only MS data is reported for **48a** and not for **48b**.

Response: Thanks for the comments and suggestions. We have conducted the EPR experiments in the absence or in the presence of alkene **2a**. Indeed, the same EPR spectra were observed. These results suggested that the initial formed sulfonyl- and carbon-centered radical intermediates **48a** ($A_N = 13.81$ G, $A_H = 18.61$ G) and **48b** ($A_N = 14.21$ G, $A_H = 20.21$ G) were captured by 5,5-dimethyl-1-pyrroline-1-oxide (DMPO). Hence, we have added the following description in the revised manuscript:

“These results strongly consistent with the EPR spin-trapping experiments using tosyl chloride (**1a**) or tert-butyl 2-iodoacetate (**1j**) as the radical precursors. Notably, regardless of the absence or presence of alkene, the same sulfonyl- or carbon-centered radical intermediates **48a** ($A_N = 13.81$ G, $A_H = 18.61$ G) and **48b** ($A_N = 14.21$ G, $A_H = 20.21$ G) were captured by 5,5-dimethyl-1-pyrroline-1-oxide (DMPO), thereby confirming the radical’s role in this pincer cobalt-catalyzed multicomponent carbonylative functionalization (Scheme 3b).”.

In addition, the MS data for **48b** is also reported in the revised Scheme 3b.

Scheme S5. EPR experiments and HR-MS analysis.

Q4) In the proposed reaction mechanism displayed in Scheme 4, the authors claim that the Co^{I} intermediate generates the radical species, thus yielding a Co^{II} intermediate, which carries out the transmetalation and CO insertion reactions. However, the stoichiometric experiments in Scheme 3d and the bottom part of Scheme 3c show that a Co^{II} species can generate the corresponding radical.

Therefore, it is possible that the initial Co^{II} complex (**A** in Scheme 4) is able to initiate the catalytic cycle, followed by the reduction of the resulting Co^{III} to Co^I complex by the arylzinc reagent. To confirm this alternative pathway, the author should carry out the EPR spin-trapping experiments in the absence of the arylzinc reactant to investigate if the first reduction step (Co^{II} to Co^I) is needed to start the catalytic cycle.

Response: Thanks for these nice suggestions.

We have performed a series of EPR spin-trapping experiments using tosyl chloride (**1a**) or *tert*-butyl 2-iodoacetate (**1j**) as the radical precursors under pincer-cobalt catalysis. Notably, both sulfonyl- or carbon-centered radical intermediates **48a** and **48b** could be observed in the presence of arylzinc pivalate **3a**. In sharp contrast, no radical signals were detected in the absence of arylzinc pivalate **3a** (Scheme S6). These observations should prove that the Co(II)-complex is not the suitable metal-source for triggering the generation of radicals, while the catalytically active Co(I)-species might be generated via further reduction or comproportionation processes of the Co(II)-complex **50**. We have added these results in the revised supporting information.

Scheme S6. EPR experiments.

Q5) In the stoichiometric experiments with the isolated Co complex **49**, the author should indicate why, in the sequential reaction, the organozinc substrate **3b** was used instead of **3a**, as in the non-sequential case.

Response: Thanks for the comments.

There is no special reason in this reaction. The only reason to use **3b** as the organozinc reagent is that the corresponding phenylmagnesium chloride is commercially available.

Q6) On page 6, lines 161-163, the authors state: "Initially, due to the lower reducibility of OPiv-supported arylzincs, the reduction of Co^{II}(bpp) **A** by stoichiometric arylzinc pivalates uniquely dictates the formation of Co^I(bpp) **B**, rather than Co⁰-complex, thereby inhibiting the generation of catalytically inert Co⁰(CO)_n." This sentence contradicts the general idea of the manuscript, which indicates that the relatively more electron-rich property of the bpp ligand prevents an overreduction to Co⁰ species. The authors should address this inconsistency.

Response: Thanks for the kind suggestions. As to this cobalt-catalyzed multicomponent carbonylative functionalization, the stereoelectronic properties of pincer ligands should be crucial to prevent the

overreduction of Co(II) complex to the catalytically inert Co(0) species by organozinc reagents (see the response for **Q1**). Moreover, we have also demonstrated that the OPiv-supported organozinc pivalates displayed appropriate effects for decreasing the reducibility of organozinc pivalates compared to the conventional halide-supported organozinc in our previous work (*Ref. 57 and 59*), thus achieving the in situ formation of catalytically active Co(I) species through a reduction of the Co(II) complex, which were confirmed by XAFS experiments. **Hence, we believe that both of the stereoelectronic property of pincer ligand and the lower reducibility of arylzinc pivalates dictate the selective formation of catalytically active Co(I) species.** We have added the new description in the revised manuscript.

Q7) The sentence on page 8, lines 182-183, needs to be revised since it does not fit within the paragraph's discussion.

Response: Many thanks for the suggestions. We have rewritten this sentence in the revised manuscript: *"Likewise, other sulfonylcarbonylated compounds 51–53 were easily within reach under the tandem catalysis using CO₂ as the initial C1 synthon."*

Q8) The authors should define the DMPO abbreviation.

Response: We have defined DMPO in the revised manuscript. *"5,5-dimethyl-1-pyrroline-1-oxide (DMPO)"*.

Q9) In Scheme 3b, for the generation of molecule **48b**, the radical electrophile **1j** was employed. The structure or name of **1j** is not mentioned anywhere in the text.

Response: We have added the structure and name of **1j** in the revised manuscript:

"These results strongly consistent with the EPR spin-trapping experiments using tosyl chloride (1a) or tert-butyl 2-iodoacetate (1j) as the radical precursors. Notably, regardless of the absence or presence of alkene, the same sulfonyl- or carbon-centered radical intermediates 48a ($A_N = 13.81$ G, $A_H = 18.61$ G) and 48b ($A_N = 14.21$ G, $A_H = 20.21$ G) were captured by 5,5-dimethyl-1-pyrroline-1-oxide (DMPO), thereby confirming the radical's role in this pincer cobalt-catalyzed multicomponent carbonylative functionalization (Scheme 3b)."

Q10) Several spelling mistakes in the main text, including in the title of the manuscript, must be revised.

Response: Many thanks for the comments and suggestions. We have carefully checked the whole manuscript and made corrections in the revised manuscript.

Reviewer #3 (Remarks to the Author):

Comments: Li, He, Luo, and co-workers demonstrated significant advancements in cobalt-catalyzed multicomponent carbonylation chemistry, demonstrating a practical and efficient method for divergent alkene carbonylative functionalization under mild conditions. Either CO or CO₂ could be utilized as C1 source in this multicomponent alkene carbonylation. With careful modulation of ligand, the toxicity of CO towards metal and side reaction generating symmetry ketones could be avoided. Mechanistic studies through the combinations of EPR, radical clock experiments, HR-MS, and DFT calculations clarify the radical nature of the reaction mechanism and provide insights into the sequence of elementary steps. I would like to strongly recommend its publications on Nature Communications with the following suggestions.

Response: We thank this reviewer for the positive comments.

Q1) The authors reported their DFT calculations on other spin states for radical-type oxidation and radical-type substitution steps in the Supplementary Information. The preference of high spin states is suggested to be discussed in the manuscript and be referred to corresponding figures in SI.

Response: We sincerely thank the reviewer for the valuable suggestions. As suggested, we have added the following description to the caption of Scheme 3: “The computed energy profiles depict the most favorable spin states of the species, while the energies of the less favorable spin states are detailed in the Supporting Information (see Figures S27 and S28).”.

Q2) The catalytic species, cobalt complex here, should be depicted on-cycle rather than off-cycle. I suggested the authors move the evolution of cobalt complex to the catalytic cycle and redraw the radical relay process as a branched pathway.

Response: Thanks for the suggestions. We have carefully redrawn the catalytic cycle in the revised manuscript (Scheme S7).

Scheme S7. Plausible catalytic cycle.

Q3) It is recommended to denote the transition state with a double dagger symbol written as a superscript instead of normal size when discussing energy barrier.

Response: Thanks for the suggestion. We have corrected this error in the revised manuscript.

Reviewer #4 (Remarks to the Author):

Comments: In this manuscript, the authors report a cobalt-catalyzed alkene carbonylation achieved through the reaction of alkenes with *p*-toluenesulfonyl chloride (TsCl) and aryl zinc reagents. Notably, other halide sources capable of initiating hydrogen atom transfer to generate radicals could also be applicable. The reaction proceeds in the presence of CoI_2 and an NNN-type ligand, with 2,6-pyrazolopyridine identified as the most effective among the ligands screened. Additionally, an investigation into the anion effect of the organozinc reagents indicates that arylzinc pivalate is the optimal choice. While related alkene carbonylation reactions using Nickel/NNN ligands have been previously reported (ACS Catal. 2023, 13, 4111–4119, J. Am. Chem. Soc. 2020, 142, 18191–18199), the innovative use of CO_2 as a CO surrogate, supported by theoretical insights and the successful late-stage functionalization of bioactive molecules, enhances the significance of this work. Therefore, the manuscript could be suitable for publication in *Nature Communications*, provided the following revisions are addressed.

Response: We thank this reviewer for the positive comments.

Q1) Is the methodology applicable to vinyl- or alkylzinc compounds instead of aryl zinc?

Response: Thanks for the comments and suggestions.

We have performed the four-component carbonylative functionalization using vinylzinc pivalates or ethylzinc pivalate as the nucleophiles under pincer-cobalt catalysis. However, no desired carbonylative functionalization of alkene was observed under the standard reaction conditions (Scheme S8).

Scheme S8. Cobalt-catalyzed carbonylative functionalization of vinylarenes with alkenyl or alkylzinc pivalates.

Q2) Is the reaction compatible with aliphatic alkenes beyond styrene analogues?

Response: Thanks for the comments. As shown in Scheme S9, aliphatic alkene, as well as various alkenes bearing a heteroatom-containing directing group have been utilized as the unsaturated substrates under the standard reaction conditions. However, no positive results for the envisioned cobalt-catalyzed four-component carbonylative functionalization reaction were detected. The corresponding alkenes remained untouched in the reaction mixture, which can be also detected by GC-analysis, only forming the symmetric diarylmethyl ketone as the byproduct.

Scheme S9. Unsuccessful substrates of olefins.

Moreover, ethylene was also used as the unsaturated substrate for the envisioned four-component carbonylative transformation. Unfortunately, no desired product was observed under 1 atm or 5 atm of ethylene and CO gaseous mixture (Scheme S10).

Scheme S10. Multicomponent carbonylation with ethylene as the unsaturated substrate.

Q3) The author proposed “Co^{II}(bpp) **A** by stoichiometric arylzinc pivalates uniquely dictates the formation of Co(bpp) **B**”---**A** is probably the complex **49**.

What happens if this complex **49** separately treated with arylzinc species? Did the author able to detect Co(bpp). The study of the mechanism of this step and detection/isolation of Co(I) species is suggested. For this particular step is there any radical inhibition.

Response: Thanks for the comments and suggestions.

As shown in Scheme S11, the reactions between complex **49** and phenylzinc pivalate **3b** under N₂ or CO atmosphere have been performed, unfortunately, the potential Co(bpp)-species could not be detected by HR-MS analysis. It is worth noting that a step-wise operation that including a mixture of stoichiometric amount of phenylzinc pivalate and Co₂(bpp) **49** under CO atmosphere at 23 °C for 30 min, then the reaction mixture was transferred into a divided reaction mixture consisting of tosyl chloride **1a** and alkene **2c** under N₂ atmosphere for another 2 h. However, no desired ketone **21** was obtained (also see the control experiments in the response to **Q5** of reviewer **4**). As such, these results should suggest that the in situ formed catalytically active Co(bpp)-species are very unstable, which may occur comproportionation process and overreduction reaction.

Scheme S11. Control experiments with Co₂(bpp) **49**.

Q4) The other cobalt precursor such as CoBr₂ was found completely inactive. What is the possible reason? Whether from **A/49** to the formation of **E** getting difficult or **A** to **B** or halogen transfer step is getting difficult. Experimental or theoretical investigations to clarify this aspect are recommended.

Response: Thanks for the comments and suggestions. Indeed, we have carefully performed the following reaction using 10 mol% of CoBr₂ instead of CoI₂ as the cobalt source. Herein, trace amount of desired product **4** was detected, while the corresponding symmetric ketone and biphenyl compound were obtained as the main byproducts (Scheme S12). Hence, these results might suggest that the halogen transfer step is getting difficult.

Scheme S12. Catalytic activity investigation of CoBr₂.

Q5) For the formation of complex **49** from CoI₂ and the ligand refluxing in the CH₃CN is required. For Scheme 3 experiment **d**, is in only 2 h at 23 °C. what happen if the reaction time is longer.

Response: Thanks for the valuable comments and suggestions.

Herein, we performed a series of control experiments via step-wise operation that including a mixture of phenylzinc pivalate and 25 mol% of CoI₂ and **bpp** (**L17**) under CO atmosphere at 23 °C for 2 min, then the reaction mixture was further transferred into a divided reaction mixture consisting of tosyl chloride **1a** and alkene **2c** under N₂ atmosphere for another 2 or 10 h, the ketone **21** (21% or 18%) was obtained in near 1:1 equiv ration to that of CoI₂-**bpp** (25%). These results demonstrated that the reaction time for the second operation under N₂ atmosphere has no influence for the yield of product **21**.

In sharp contrast, a mixture of phenylzinc pivalate and 25 mol% of CoI₂ and **bpp** (**L17**) under CO atmosphere at 23 °C for 30 min, then the reaction mixture was further transferred into a divided reaction mixture consisting of tosyl chloride **1a** and alkene **2c** under N₂ atmosphere for another 2 h, no desired product **21** was observed. These results suggested that a longer reaction time in the latter radical relay coupling step has no influence to the yield of product **21**, while a short reaction time in the former step is crucial important for the in situ formation of catalytically active acyl-cobalt species, which might be ascribed to the highly unstable of Co(I)-species (Scheme S13).

Scheme S13. Investigation of reaction time through step-wise operation.

Q6) It is surprising that the authors only screened *N,N,N*-ligands, especially since the yields for several compounds (e.g., **10**, **19**, **20**, **30**) are moderate. Why were PNP cobalt complexes not explored as potential catalysts for this transformation?

Response: Thanks for the suggestions.

A representative *PNP*-type ligand was used for the following reactions. Unfortunately, no desired products were detected (Scheme S14). These results might suggest that the *PNP*-type ligands are not suitable for the present reaction. Indeed, the relatively lower yields of products (**19**, **20**, **30**) under the standard reaction conditions might be ascribed to the steric hindrance of the arylzinc reagent or alkene.

Scheme S14. Investigation of *PNP*-type ligand.

Q7) A dioxane-coordinated complex was detected by HRMS. It may be more appropriate to base the theoretical calculations on this complex; however, further input from a theoretical expert could help determine the most suitable approach.

Response: Thanks for the comments and suggestions.

Scheme S15. EPR experiments.

We have performed a series of EPR spin-trapping experiments using tosyl chloride (**1a**) or *tert*-butyl 2-iodoacetate (**1j**) as the radical precursors under pincer-cobalt catalysis. Notably, both sulfonyl- or carbon-centered radical intermediates **48a** and **48b** could be observed in the presence of arylzinc pivalate **3a**. In sharp contrast, no radical signals were detected in the absence of arylzinc pivalate **3a** (Scheme S15). **These observations should prove that the Co(II)-complex 49 or 50 are not the suitable metal-source for triggering the generation of radicals**, while the catalytically active Co(I)-

species might be generated via further reduction or comproportionation processes of the Co(II)-complex 50.

Q8) In the theoretical pathway from ⁴II to ³IV, is there a possibility of forming a Co(II) intermediate through the dissociation of species I? This alternative route involving Co(II), rather than Co(III), might represent a lower-energy pathway.

Response: We sincerely thank the reviewer for the valuable comments and suggestions. We examined the possible dissociation pathways of the iodine species based on species ³IV (Scheme S16). The dissociation of the iodine radical to form a Co(II) species is highly endothermic (43.6 kcal/mol), and the dissociation of the iodide anion to yield a Co(III) cationic species is similarly unfavorable (33.5 kcal/mol). Thus, the energy profiles presented in the manuscript more appropriately reflect the plausible reaction pathway.

Scheme S16. DFT experiments.

Attached you will find a revised manuscript and supporting information.

With best regards

Jie Li